# An In-Depth Study on Deep Learning Model Cloning

**Bin Hu** [1]   **Xiancong Pan** [1]   **Dongjin Yu** [1]   **Tianyi Hu** [1]

## Abstract

Artificial intelligence has achieved remarkable breakthroughs in fields such as text, image, and video analysis, with deep learning serving as the mainstream paradigm. Trained deep learning models can be integrated into various applications either through fine-tuning or without any modification. While this practice promotes the advancement of artificial intelligence, it also raises concerns regarding intellectual property protection and information security risks. Therefore, it is necessary to propose relevant methods to measure the similarity between models. Existing code clone detection techniques are insufficient to address this issue. In this paper, we provide the first definition of model cloning and design a method for model similarity detection. The framework characterizes model topology at the structural level based on normalized computational graphs, and at the weight level, it employs a method that does not require explicit parameter alignment to measure the statistical similarity of weight parameters. Experiments on a synthetic model clone benchmark dataset and real-world open-source models demonstrate that the proposed method can accurately detect similar models. This method provides a unified and extensible quantitative foundation for model lineage analysis, model retrieval, and intellectual property protection of models.

## 1. Introduction

With the continued advancement of artificial neural networks, artificial intelligence has achieved major breakthroughs in recent years. Deep learning models—including large language models and visual diffusion models—have been widely adopted in applications such as machine trans-

lation, conversational systems, and face recognition (Zhao et al., 2023; Yang et al., 2023). In particular, the emergence of large language models (Radford et al., 2018; 2019; Brown et al., 2020; Achiam et al., 2023) has substantially expanded model capability, application scope, and societal impact, making deep learning the primary driver of modern AI systems. Training and tuning such models require substantial computational resources and human effort, which are often inaccessible to many teams. As a result, model reuse, transfer, and derivation have become the dominant paradigm for integrating pretrained models into downstream applications. While this paradigm accelerates innovation and deployment, large-scale model reuse also introduces significant risks, including intellectual property conflicts and security or compliance concerns (Bender et al., 2021). According to our statistics, as of December 2025, Hugging-Face hosts over 2.3 million open-weight models, yet fewer than 30% explicitly specify license information. This highlights the prevalence of unclear ownership and provenance in the open-source model ecosystem, making it difficult for users to assess copyright status, evolutionary lineage, and potential security risks. These challenges motivate the need for effective methods to quantitatively measure similarity between deep learning models, supporting model provenance analysis and risk identification.

In software engineering, a rich body of work has addressed software and code similarity, with techniques ranging from syntactic analysis to deep learning–based semantic embeddings (Roy & Cordy, 2008; Sajnani et al., 2016; Svajlenko & Roy, 2017; Wang et al., 2020; Hua et al., 2020; Vijayanandan et al., 2025). Although highly effective for code clone detection, these methods are not directly applicable to deep learning models, which consist of interconnected neural components linked by directed, weighted edges and naturally form complex computation graphs.

In this work, we transfer core ideas from code clone detection to the domain of deep learning models and propose MCDetector, a model similarity and clone detection method tailored to deep learning systems. MCDetector treats a model as a comparable entity jointly characterized by its computational structure and parameter distribution. At the structural level, models are transformed into a unified computation graph representation, and SimHash (Manku et al., 2007) is used to extract structural fingerprints captur-

---

[1]School of Computer Science and Technology, Hangzhou Dianzi University, Hangzhou, China. Correspondence to: Dongjin Yu <yudj@hdu.edu.cn>.

*Proceedings of the $43^{rd}$ International Conference on Machine Learning*, Seoul, South Korea. PMLR 306, 2026. Copyright 2026 by the author(s).

ing operator-level topology. At the parameter level, trainable weights are modeled as tensor sets, and bidirectional maximum cosine matching is employed to measure similarity between parameter direction distributions (Min & Wang, 2025). MCDetector requires no additional training, is framework-agnostic, and supports cross-framework model similarity analysis. Experiments on both synthetic datasets and real open-source models demonstrate its effectiveness in model clone detection scenarios.

The main contributions of this work are summarized as follows:

- We introduce the concept of deep learning model clones and define four distinct clone types. We further construct **ModelCloneBench**[1], a synthetic dataset with explicit Type-1, Type-2 and Type-3 clone labels, providing a controlled and reproducible benchmark.

- We propose **MCDetector**, a framework-agnostic whitebox method that jointly measures structural and weight similarity to detect Type-1, Type-2 and Type-3 model clones.

- We conduct large-scale experiments on both synthetic and real open-source models, systematically evaluating the effectiveness of the proposed method across clone categories.

The remainder of this paper is organized as follows. Section 2 introduces the background and formal definitions of model clones. Section 3 presents the proposed detection method. Section 4 describes datasets, experimental settings, and results. Section 5 discusses threats to validity, Section 6 reviews related work, and Section 7 concludes the paper.

## 2. Motivation and Background

### 2.1. Motivation

In practical intelligent software engineering, most deep learning models are not trained from scratch, but are derived from pretrained foundation models through fine-tuning (Howard & Ruder, 2018), distillation (Hinton et al., 2015), parameter-efficient adaptation (Hu et al., 2022), or structural compression (Han et al., 2015). While this paradigm significantly improves development efficiency and reduces cost, it also introduces a large number of models with highly similar structures or overlapping functionality into open-source communities and related software ecosystems.

Figure 1 shows several closely related models from the HuggingFace community. *bert-base-uncased*, released in 2018, is built on a Transformer encoder architecture. *roberta-base* preserves the same network structure while optimizing the

___
[1]https://github.com/nikoHu/ModelCloneBench

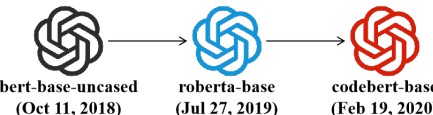

bert-base-uncased    roberta-base    codebert-base
(Oct 11, 2018)    (Jul 27, 2019)    (Feb 19, 2020)

*Figure 1.* An example of model derivation in open-source ecosystems.

pretraining strategy, and *codebert-base* further extends this architecture by incorporating source code data and code–text alignment tasks. Although maintained by different organizations and distributed under licenses such as Apache-2.0 and MIT, these models clearly evolve from a common architectural lineage and exhibit strong structural inheritance. This pattern is widespread in modern model repositories. Simple keyword searches on HuggingFace return 3,454 models containing "Transformer", 46,872 containing "BERT", and 5,595 containing "Diffusion". Although naming similarity does not necessarily imply substantive equivalence, these numbers highlight the prevalence of models that are closely related in structure, parameters, or functionality.

The large-scale proliferation of model clones and closely related variants, while accelerating adoption, also introduces significant risks. Similarity between models can amplify intellectual property and compliance concerns, particularly in commercial settings where the boundaries of derived models are unclear. Security vulnerabilities may also propagate: flaws or backdoors in a base model are likely to be inherited by its derivatives. Moreover, unclear model provenance undermines trustworthiness and traceability, complicating reliability assessment and accountability in high-stakes domains such as healthcare and finance.

These challenges underscore the need for principled methods to detect model similarity and trace model provenance. Such capabilities are essential not only for understanding the evolutionary structure of open-source model ecosystems, but also for supporting secure deployment, compliant usage, and sustainable innovation.

### 2.2. Background

#### 2.2.1. MODEL CLONE

Deep learning models are composed of artificial neural networks and can be concretely viewed as collections of interconnected operator nodes, where different connections are associated with different parameter values. If a trained model is reused without any modification and is released as a new model by merely changing its name, the two models are identical in both structure and weights. If a trained model is fine-tuned on a new domain-specific dataset to produce a new model, the resulting model shares the same network structure, while the connection weights between nodes may differ. If several mature neural networks are

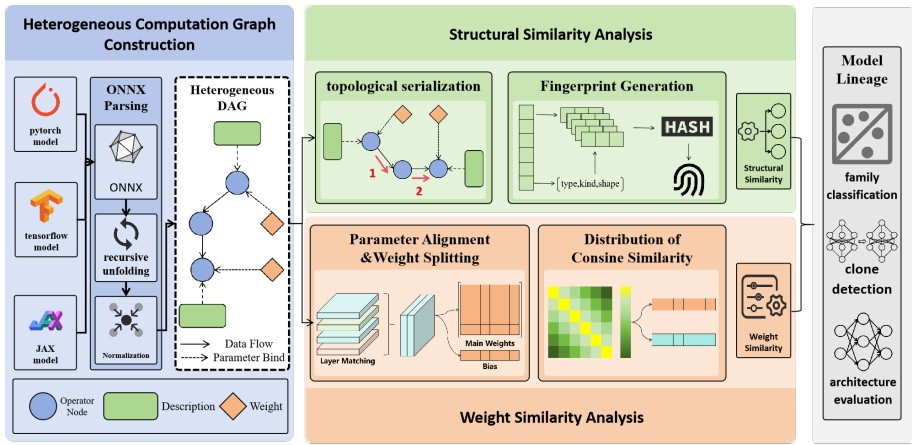

*Figure 2.* Overview of MCDetector

combined or spliced to construct a new neural network model, the new model exhibits partial structural similarity with the original models from which it is derived. Accordingly, given two models $M^{(A)} = (V^{(A)}, E^{(A)}, \Theta^{(A)})$ and $M^{(B)} = (V^{(B)}, E^{(B)}, \Theta^{(B)})$, where $V$ denotes the set of computation nodes, $E \subseteq V \times V$ represents tensor dependency relations between nodes, and $\Theta = \{\theta_1, \theta_2, \ldots, \theta_{|\Theta|}\}$ denotes the set of all learnable parameters in the model, we define deep learning model clones as follows:

**Type-1 Model Clones:** If two models are identical in both structure and parameters, we classify them as Type-1 Model Clone, formally defined as $(V^{(A)}, E^{(A)}) \cong (V^{(B)}, E^{(B)}), \quad \Theta^{(A)} = \Theta^{(B)}$

**Type-2 Model Clones:** If two models share an identical structure but differ in their parameters, they are classified as Type-2 Model Clones, defined as $(V^{(A)}, E^{(A)}) \cong (V^{(B)}, E^{(B)}), \quad \Theta^{(A)} \sim \Theta^{(B)}$

**Type-3 Model Clones:** If two models differ in structure, but their overall structural similarity exceeds a predefined threshold and their parameter distributions remain highly consistent, they are classified as Type-3 Model Clones, defined as $(V^{(A)}, E^{(A)}) \not\cong (V^{(B)}, E^{(B)}), S_{\text{struct}}(M^{(A)}, M^{(B)}) \geq \tau_s, \quad \Theta^{(A)} \sim \Theta^{(B)}$

**Type-4 Model Clones:** If two models differ in network structure but exhibit the same functionality—for example, an RNN-based translation model and a Transformer-based translation model that perform the same translation task with highly similar input-output behavior—then the two models are classified as Type-4 model clones, defined as $(V^{(A)}, E^{(A)}) \not\cong (V^{(B)}, E^{(B)}), \quad S_{\text{func}}(f_A, f_B) \geq \tau_f$. where $f_A(i)$ denotes the output of model A with input i. Detecting Type-4 behavioral clones requires output-level or representation-level analysis and is beyond the scope of MCDetector.

## 3. Approach

To detect Type-1 to Type-3 model clones defined above, we design a framework-agnostic model clone detection method, termed MCDetector. MCDetector measures similarity between deep learning models from two complementary perspectives: structure and weights. At the structural level, it identifies models with similar architectures by comparing network topologies; at the weight level, it assesses consistency in parameter space to quantify the degree of parameter similarity. As illustrated in Figure 2, our pipeline consists of three stages:

- **Unified model representation:** construct a framework-independent heterogeneous computation graph based on the ONNX intermediate representation;

- **Structural similarity analysis:** compute structural similarity $S_{\text{struct}}$ via stable topological serialization and SimHash-based fingerprints;

- **Weight similarity analysis:** compute weight similarity $S_{\text{weight}}$ using the SDOCS metric based on bidirectional maximum cosine distributions.

### 3.1. Unified Model Representation

In practical applications, models are often built using different software frameworks, such as TensorFlow, PyTorch, Keras, or PaddlePaddle. These frameworks differ substantially in how they define computation graphs, implement operators, store weights, and manage runtime environments, resulting in binary-level incompatibilities between models. If a detection method relies on the native format of a specific framework, its applicability becomes severely limited, making it difficult to perform scalable and consistent analysis across heterogeneous model ecosystems. To address this challenge, we adopt ONNX (Open Neural Network Exchange) as a unified intermediate representation. ONNX is

an open standard for representing machine learning models, designed to enable interoperability across frameworks, tools, and hardware platforms(onn, 2017). It defines a common computation graph representation, operator set, and data types, allowing models from diverse frameworks to be converted into a unified format that abstracts away framework-specific details while preserving the semantic information of model structure and parameters.

**Graph Abstraction Based on ONNX:** Under the ONNX representation, we abstract a model as a heterogeneous directed acyclic graph $G = (V, E)$, where $V$ is the set of nodes, and each node corresponds to an operator (e.g., Conv, MatMul, Attention). Each node is associated with dimensional attributes. For example, consider two linear operators: one maps an input of size $2 \times 10$ to an output of size $10 \times 1$, while the other maps $3 \times 5$ to $5 \times 2$. Although both are linear layers, they are treated as structurally distinct due to their different dimensional characteristics. In practice, if a dimension extracted from ONNX is missing or symbolic (e.g., $unk\_*$), we normalize it using a unified placeholder token $< VAR >$. $E$ is the set of edges, where each edge represents the flow of tensors between operators, including both input and output tensors.

### 3.2. Structural Similarity Analysis

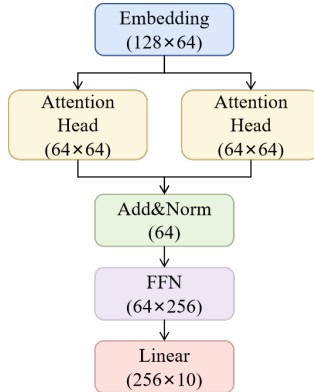

*Figure 3.* Structural serialization example

Directly comparing neural computation graphs using exact graph matching or graph edit distance is difficult to scale, as GED is NP-hard and recent ILP-based GED solvers still require minutes on large graph benchmarks such as CORA and PUBMED (D'Ascenzo et al., 2025). To address this issue, MCDetector exploits the directed data-flow nature of neural network computation graphs. We serialize each graph into a deterministic topological sequence of operator-shape tokens and apply SimHash to obtain a compact structural fingerprint. This reduces structural comparison from expensive graph matching to efficient approximate sequence comparison.

This process can be formalized as a mapping $f : G \rightarrow$

$S = [v_1, v_2, \ldots, v_n]$ where the sequence $S$ depends solely on the model structure and is independent of node naming.Each node $v_i$ is embedded as a tuple $\mathbf{v}_i = [\text{type}(v_i), \text{shape}(v_i)]$ where type denotes the operator type and shape captures the tensor dimensional characteristics.

For network nodes with explicit input–output dependencies, we traverse them in topological order. When a layer contains a multi-head structure, each head is treated as an individual node, and all heads are unfolded into a sequential list. As illustrated in Figure 3, consider a simplified attention-based module in which a $128 \times 64$ embedding layer is followed by two $64 \times 64$ attention heads, an Add&Norm node, an FFN node, and a final linear layer. After serialization, the corresponding node sequence is:$[[\text{Embedding}, 128 \times 64], [\text{Attention Head}, 64 \times 64], [\text{Attention Head}, 64 \times 64], [\text{Add\&Norm}, 64], [\text{FFN}, 64 \times 256], [\text{Linear}, 256 \times 10]]$. We then construct n-gram fragments of length $n$ over the node sequence:$\mathcal{G}_k = (\mathbf{s}_k, \mathbf{s}_{k+1}, \ldots, \mathbf{s}_{k+n-1})$and apply linear congruential expansion to generate a SimHash fingerprint for the sequence. Let the resulting SimHash values for models $A$ and $B$ be $h_A$ and $h_B$, respectively. The structural similarity between the two models is computed as:

$$\text{Sim}_{\text{struct}}(A, B) = 1 - \frac{\text{Hamming}(h_A, h_B)}{d}$$

,where $\text{Hamming}(h_A, h_B)$ denotes the Hamming distance between the two fingerprints.

### 3.3. Weight Similarity Analysis

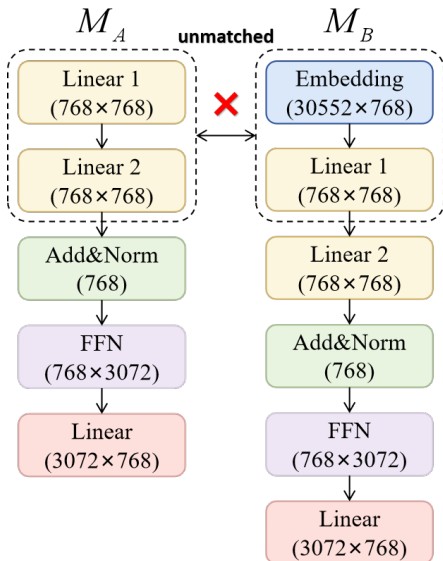

*Figure 4.* Layer alignment for weight similarity

Based on the serialized network structure sequences obtained from structural comparison, we adopt a common-substring–based matching strategy to identify and extract

locally consistent structural components shared by two models. These aligned components are then used as the basis for fine-grained weight similarity analysis. The core idea is to perform local pattern matching at the sequence level using a sliding window, thereby capturing structural correspondences between models in terms of both operator types and parameter shapes. Given two models $M_A$ and $M_B$, we first derive their sequence representations according to the topological order of their computation graphs: $S^{(A)} = [v_1^{(A)}, v_2^{(A)}, \ldots, v_{n_A}^{(A)}]$, $S^{(B)} = [v_1^{(B)}, v_2^{(B)}, \ldots, v_{n_B}^{(B)}]$. To extract structurally consistent contiguous layer fragments, we apply a sliding window of fixed size 2 over $S^{(A)}$, advancing one node at a time. Each window is defined as a pair of adjacent nodes: $W_i^{(A)} = (v_i^{(A)}, v_{i+1}^{(A)})$, $W_j^{(B)} = (v_j^{(B)}, v_{j+1}^{(B)})$. For each window $W_i^{(A)}$ in $S^{(A)}$, we slide over $S^{(B)}$ to search for a potentially matching window $W_j^{(B)}$. Two windows are considered matchable if and only if all of the following conditions are satisfied:

$$W_i^{(A)} \sim W_j^{(B)} \iff \begin{aligned} & \operatorname{op}(v_i^{(A)}) = \operatorname{op}(v_j^{(B)}) \\ \wedge\ & \operatorname{op}(v_{i+1}^{(A)}) = \operatorname{op}(v_{j+1}^{(B)}) \\ \wedge\ & \operatorname{shape}(v_i^{(A)}) = \operatorname{shape}(v_j^{(B)}) \\ \wedge\ & \operatorname{shape}(v_{i+1}^{(A)}) = \operatorname{shape}(v_{j+1}^{(B)}) \end{aligned}$$

. Once a matching position in $S^{(B)}$ is found, the corresponding node pairs within the window are added to the layer-matching set $P$. The window then advances by one node in $S^{(A)}$, and the matching process is repeated until the entire sequence has been traversed. In the subsequent analysis, weight similarity is computed only for the matched layer pairs, while unmatched nodes are automatically excluded. As shown in Figure 4, we apply a sliding window with size 2 to perform local matching over the serialized layer representations of two models. The first window of model $M_A$ is: $A_1 = [\operatorname{Linear}(768 \times 768), \operatorname{Linear}(768 \times 768)]$. In contrast, the first window of model $M_B$ is: $B_1 = [\operatorname{Embedding}(30522 \times 768), \operatorname{Linear}(768 \times 768)]$. Since $B_1$ contains an additional Embedding layer, its operator types and parameter shapes cannot be fully matched with those of $A_1$. Therefore, this window is not aligned. The window then slides forward along the sequence of $M_B$. When it reaches: $B_2 = [\operatorname{Linear}(768 \times 768), \operatorname{Linear}(768 \times 768)]$, $B_2$ and $A_1$ are consistent in both operator types and parameter shapes, and are therefore regarded as a successfully matched local structural fragment. By repeating this process, multiple alignments between consecutive nodes can be established across the two models, forming structurally corresponding local subgraph fragments. After obtaining structurally matched layer pairs, the parameters of each layer are further divided into two components for analysis: the primary weight block (W), corresponding to multi-dimensional matrix parameters (e.g., convolution kernels or projection matrices), and the bias block (b), corresponding

to one-dimensional bias vectors, which appear only in layer types that support biases.

For a matched layer pair in the layer-matching set $P$, let the corresponding parameter matrices be $W_A$ and $W_B$. We first perform column-wise vectorization and normalization, and then define the bidirectional maximum cosine similarity statistics as:

$$s_X = \frac{1}{m_A} \sum_{i=1}^{m_A} \max_j |\langle x_i^{(A)}, x_j^{(B)} \rangle|,$$

$$s_Y = \frac{1}{m_B} \sum_{j=1}^{m_B} \max_i |\langle x_j^{(B)}, x_i^{(A)} \rangle|$$

The layer-level similarity is defined as

$$\operatorname{SDOCS}(L^{(A)}, L^{(B)}) = \frac{1}{2}(s_X + s_Y).$$

This bidirectional maximum-matching mechanism reduces sensitivity to channel permutations, feature reordering, and sparsification perturbations. As illustrated in Figure 4, consider the matched weight matrices $W_1^{(A)}$ and $W_1^{(B)}$ of $Linear_1$ in models $M_A$ and $M_B$, respectively. For illustration, we treat $W_1^{(A)}$ as a set of column vectors $\{x_1^{(A)}, x_2^{(A)}, \ldots, x_{m_A}^{(A)}\}$ and similarly represent the corresponding matched weight matrix as $\{x_1^{(B)}, x_2^{(B)}, \ldots, x_{m_B}^{(B)}\}$. For each column vector $x_i^{(A)}$ in $W_1^{(A)}$, we compute its absolute cosine similarity with all column vectors in $W_1^{(B)}$, and select the maximum value as its best-matched similarity. Averaging these maximum values over all column vectors in $W_1^{(A)}$ gives $s_X$, and applying the same procedure in the reverse direction gives $s_Y$. Together, $s_X$ and $s_Y$ characterize the parameter-distribution similarity between $W_1^{(A)}$ and $W_1^{(B)}$.

The global weight similarity is computed via a layer-wise weighted average:

$$SDOCS_{\text{weight}}^{\text{global}} = \frac{\sum_k n_k SDOCS_{\text{weight}}^k}{\sum_k n_k},$$

where $n_k$ is the effective parameter size of the $k$-th matched layer.

For each model pair, this process yields a two-dimensional similarity vector $(S_{\text{struct}}, S_{\text{weight}})$, where $S_{\text{struct}}$ captures network-topology similarity and $S_{\text{weight}}$ captures weight-parameter similarity.

## 4. Experiments

In this section, we evaluate MCDetector from two perspectives. First, we use a synthetic model-clone benchmark with explicit Type-1 to Type-3 labels to examine whether MCDetector can distinguish different clone types and to analyze

the respective roles of structural and weight similarity. Second, we evaluate MCDetector on real-world open-source model families from HuggingFace to examine whether it can reveal similarity patterns and retrieve documented related models in realistic model ecosystems. We formulate the following research questions:

- **RQ1:** Can MCDetector effectively distinguish Type-1 to Type-3 model clones, and what roles do structural and weight similarity play?

- **RQ2:** Can MCDetector reveal and retrieve documented model relationships in real-world open-source model ecosystems?

### 4.1. Datasets

We additionally construct a synthetic deep learning model clone dataset, termed ModelCloneBench. This dataset is built from a variety of base neural network components, including linear models, MLP layers, convolutional kernels, and lightweight Transformer architectures. For each base component, we construct baseline networks by stacking different numbers of layers, and then generate model variants through a series of controlled transformations to cover three clone types: Type-1: the original model structure and parameters are preserved without any modification, serving as positive reference samples for clone detection; Type-2: the network structure is kept unchanged, while model parameters are altered through random initialization, partial weight perturbation, or retraining on different data; Type-3: moderate architectural changes are introduced on top of the original structure, such as adding or removing layers, changing convolution kernel sizes, or modifying the number of attention heads, thereby introducing structural differences while maintaining functional similarity. The final dataset contains a total of 208 model clones, including 36 Type-1, 90 Type-2, and 82 Type-3 samples.

To evaluate the applicability of MCDetector in real-world model ecosystems, we further collect a representative set of open-source model families from the HuggingFace platform. The selected models cover mainstream Transformer architectures and exhibit explicit reuse, fine-tuning, and derivation relationships. Specifically, we select six base models from four major families: Meta's LLaMA3.2-1B and LLaMA3.2-3B, Alibaba's Qwen2.5-3B and Qwen2.5-7B, Microsoft's Phi-4-mini, and Google's Gemma3-1B.For each base model, we further gather multiple real-world derivatives, including instruction-tuned models, adapter-based fine-tuned models, as well as model compositions or lightweight modified variants. This dataset reflects the natural distribution of model reuse and derivation in real open-source ecosystems, and is primarily used to assess the discriminative capability and global similarity patterns of the proposed method in realistic settings.

*Table 1.* Clone detection performance by clone type

| Clone Type | Precision | Recall | F1-score |
|------------|-----------|--------|----------|
| Type-1 | 1.0000 | 1.0000 | 1.0000 |
| Type-2 | 1.0000 | 1.0000 | 1.0000 |
| Type-3 | 1.0000 | 0.9146 | 0.9554 |

### 4.2. RQ1: Clone-Type Detection and Component Analysis

#### 4.2.1. OVERALL CLONE-TYPE DETECTION

We first evaluate the proposed method on the synthetic model-clone dataset to assess its ability to distinguish different clone types. In this experiment, the structural similarity threshold for Type-3 clones is set to $\tau_s = 0.7$, based on the threshold sensitivity analysis reported in Appendix A. As shown in Table 1, the method achieves very high accuracy for Type-1 and Type-2 clones. In these cases, models remain structurally identical, with parameters differing only through equivalent transformations or continuous evolution, allowing all Type-1 and Type-2 clone pairs to be correctly identified. For Type-3 clones, a small number of false negatives are observed. Further inspection reveals that these cases involve substantial structural derivations, where significant topological changes lead to reduced structural similarity that no longer meets the clone-identification criterion.

To further examine distributional differences among clone types, we analyze the two-dimensional similarity vectors of model pairs, as shown in Figure 6. The x-axis denotes structural similarity and the y-axis denotes weight similarity, with different markers indicating different clone types. The results show clearly separable regions in the similarity space: Type-1 clones cluster in the upper-right region with both similarities close to 1.0; Type-2 clones retain high structural similarity but exhibit greater variation in weight similarity; and Type-3 clones mainly appear in regions with lower structural similarity and more dispersed weight similarity. These observations highlight the complementarity of structural and weight similarity and demonstrate the effectiveness of their joint analysis for distinguishing clone types.

#### 4.2.2. COMPONENT ANALYSIS OF STRUCTURAL AND WEIGHT SIMILARITY

To examine whether structural and weight similarity play different roles, we conduct a component analysis on the synthetic benchmark. The results are shown in Table 2. Structural similarity alone can identify Type-3 clones, but it cannot distinguish Type-1 from Type-2 because both have identical structures. Weight similarity alone is also insufficient for complete clone-type, because Type-1 and Type-2 require structural consistency by definition. Therefore, the

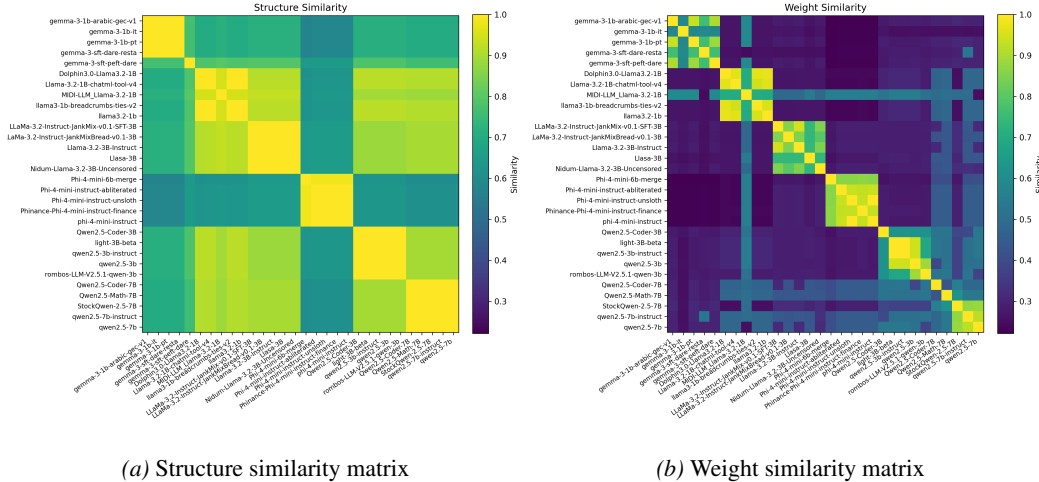

*(a)* Structure similarity matrix        *(b)* Weight similarity matrix

*Figure 5.* Joint analysis of structural and weight similarity in real open-source model families

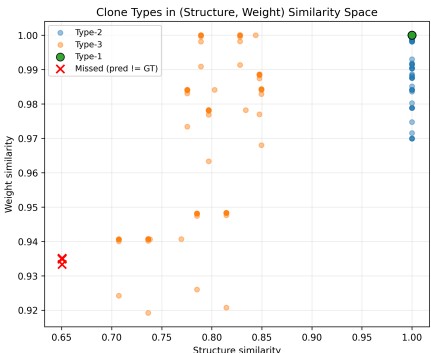

*Figure 6.* Distribution of different clone types in the joint structural–weight similarity space

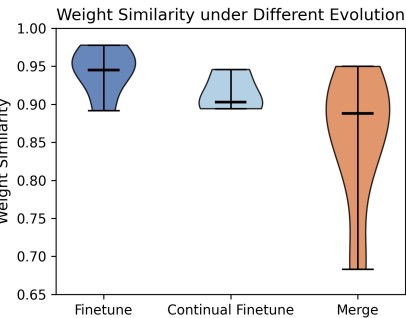

*Figure 7.* Weight similarity distributions under different model evolution operations

two components are complementary rather than redundant. In addition, we compare the proposed bidirectional maximum cosine matching with direct cosine similarity and Frobenius-based similarity in Appendix B, further validating the effectiveness of the weight similarity component.

**Answer to RQ1:** MCDetector accurately distinguishes Type-1 to Type-3 clones on the synthetic benchmark. Structural similarity and weight similarity are complementary: structural similarity identifies structurally modified Type-3 clones, while weight similarity further distinguishes exact copies from parameter-evolved variants under identical structures.

### 4.3. RQ2: Real-World Model Relationships and Retrieval

#### 4.3.1. SIMILARITY MATRIX ANALYSIS ON OPEN-SOURCE MODEL FAMILIES

We apply MCDetector to a real open-source model ecosystem to evaluate its behavior in a realistic setting where ex-

plicit clone labels are unavailable and complex derivation relationships coexist. Rather than relying on manual annotations, we assess generalization and discriminative stability through similarity distribution patterns and intra-family consistency. Figure 5 reports the results.

Figure 5a shows the structural similarity matrix, and Figure 5b shows the corresponding weight similarity matrix. Axes correspond to different model instances, and color intensity encodes similarity, with lighter colors indicating higher similarity. Regions near the diagonal are consistently lighter, while off-diagonal regions are darker, revealing clear and stable low-similarity boundaries between different model families. Models within the same family form distinct high-similarity blocks, whereas cross-family pairs exhibit uniformly low structural similarity. These results indicate that structural similarity effectively captures shared architectural and computational-topology characteristics in real-world settings. For structurally similar models, weight similarity further exposes differences in parameter distributions, providing additional discriminative power.

*Table 2.* Ablation study on structural and weight similarity.

| Clone Type | MCDetector | | | Only Structure | | | Only Weight | | |
|---|---|---|---|---|---|---|---|---|---|
| | P | R | F1 | P | R | F1 | P | R | F1 |
| Type-1 | 1.00 | 1.00 | 1.00 | N/A | N/A | N/A | N/A | N/A | N/A |
| Type-2 | 1.00 | 1.00 | 1.00 | N/A | N/A | N/A | N/A | N/A | N/A |
| Type-3 | 1.00 | 0.91 | 0.96 | 1.00 | 0.91 | 0.96 | N/A | N/A | N/A |

### 4.3.2. RETRIEVAL EVALUATION BASED ON DOCUMENTED DERIVATION METADATA

To provide a quantitative evaluation on real-world models, we use the `base_model` field and documented derivation chains in HuggingFace model cards as weak supervision. Given a model, we compute its similarity to other models using our proposed MCDetector to retrieve models with derivation relationships. The computed similarities are ranked in descending order, and a retrieval is considered successful if at least one relevant model appears in the top-k results (Hit@k).

*Table 3.* Retrieval performance on documented model derivation relationships

| Approach | Hit@1 | Hit@2 | Hit@3 |
|---|---|---|---|
| MCDetector | 0.58 | 0.92 | 1.00 |

As shown in Table 3, MCDetector retrieves a documented related model within the top-3 results for all queries. The lower Hit@1 mainly comes from cases involving substantial fine-tuning, instruction tuning, or model merging, where weight similarity to the `base_model` decreases. These results indicate that the combination of structural and weight similarity is useful for prioritizing related models in open-source model ecosystems.

### 4.3.3. EVOLUTIONARY TRANSFORMATION ANALYSIS

We systematically aggregate and analyze the weight similarity between model pairs along the evolution chains and visualize the results using violin plots. As shown in Figure 7, the x-axis denotes different types of derivation relationships between model pairs, while the y-axis represents their weight similarity. We observe that directly fine-tuned models exhibit consistently high weight similarity with their source models, indicating substantial inheritance of the original parameter structure. In contrast, continuously fine-tuned models show lower weight similarity than directly fine-tuned models, with their distributions shifting toward smaller values, reflecting the gradual weakening of parameter inheritance across multiple fine-tuning stages.In sharp contrast to these fine-tuning paths, merged models display substantially lower weight similarity overall, accompanied

by a much wider distribution range, indicating significantly greater dispersion.

**Answer to RQ2:** MCDetector reveals significant similarity difference among model families in the open-source ecosystem, it retrieves related models with Hit@3 of 1.00. Structural similarity captures family-level relationships, while weight similarity reflects parameter-level changes caused by fine-tuning, continuous fine-tuning, and model merging.

## 5. Threats to Validity

### 5.1. Internal Validity

Our evaluation combines a synthetic model-clone dataset with real open-source models. While the synthetic dataset provides explicit clone-type labels for controlled analysis, its construction inevitably simplifies the complexity of real-world model evolution, and some boundary cases may be underrepresented. For real models, derivation relationships are inferred from publicly available information rather than exhaustive manual annotation, so the analysis emphasizes distributional consistency and interpretability rather than strict classification accuracy. These factors may limit the granularity of the conclusions.

### 5.2. External Validity

Our experiments focus on Transformer-based language models and common evolution paths, including fine-tuning, continuous fine-tuning, and model merging. Although representative, extending the method to other architectures, such as multimodal models, requires further validation. In addition, the approach depends on ONNX as an intermediate representation; models that cannot be fully exported to ONNX may limit its applicability.

MCDetector is a white-box method and requires access to model structures and parameters. It is therefore not directly applicable to encrypted models, closed-source models, or black-box API services. In addition, the real-world retrieval evaluation relies on HuggingFace model-card metadata, which may be incomplete, inconsistent, or noisy.

# 6. Related Work

## 6.1. Code Clone Research in Software Engineering

Code clones refer to code fragments that are highly similar at the syntactic, structural, or semantic level. Prior studies have systematically established clone definitions, taxonomies, and research directions, showing that while cloning facilitates reuse and development efficiency, it also introduces long-term risks such as defect propagation and increased maintenance cost. To analyze clone distribution and evolution, researchers proposed metrics including clone density, entropy, change dispersion, and clone genealogies, which capture stability and risk patterns over long-term software evolution (Hu et al., 2025; Mondal et al., 2014; 2016; 2015; Wagner et al., 2016; Barbour et al., 2018). Extensive empirical evidence further shows that defects often propagate through copy-and-modify processes, especially in near-miss and inconsistent clones (Mondal et al., 2017; 2019; Islam et al., 2019; Aversano & Nardi, 2019).

To detect different clone types, a broad spectrum of techniques has been developed, including token-based, AST-based, PDG-based, and deep learning–based approaches (Roy & Cordy, 2008; Sajnani et al., 2016; Svajlenko & Roy, 2017; Jiang et al., 2007; Wang et al., 2017; 2020; Hua et al., 2020; Vijayanandan et al., 2025; Inoue & Higo, 2024). After more than two decades, code clone research has matured into a well-established methodology centered on abstracting programs into structured representations and analyzing similarity and evolution from structural, semantic, and temporal perspectives. This paradigm provides a solid methodological foundation for extending clone analysis to deep learning models.

## 6.2. From Code Clones to Model Clones

With the widespread adoption of pretraining, fine-tuning, distillation, and structural compression, deep learning models increasingly exhibit complex derivation and evolution patterns, akin to copy-and-variant processes in code cloning. However, most existing studies analyze model similarity from a single perspective—such as structure or parameters—lacking a unified abstraction for modeling clone relationships.

At the structural level, neural networks can be represented as computational graphs. NAS-Bench-101 formalizes convolutional networks as DAGs and supports reproducible architecture comparison (Ying et al., 2019). More general graph similarity methods include graph edit distance (GED) (Riesen & Bunke, 2009) and Weisfeiler–Lehman (WL) graph kernels (Shervashidze et al., 2011). However, their high computational cost and limited scalability restrict their applicability to large, heterogeneous model ecosystems.

At the parameter level, Model merging and averaging studies reveal that fine-tuned models often reside in shared low-loss basins (Wortsman et al., 2022). Due to neuron permutation symmetry, direct parameter-wise comparison is challenging. Approaches such as Git Re-Basin address this by explicit neuron alignment (Ainsworth et al., 2022), while related work explores Bayesian and Fisher-information-based formulations (Matena & Raffel, 2022; Yurochkin et al., 2019). To avoid expensive alignment, DOCS characterizes similarity via the distribution of cosine similarities across weights, yielding robustness to local perturbations (Min & Wang, 2025).

Representation-level similarity has also been widely studied. SVCCA (Raghu et al., 2017) and CKA (Kornblith et al., 2019) compare layer-wise activations to analyze shared representations and training dynamics (Morcos et al., 2018; Bansal et al., 2021). More recent work proposes gradient-based model fingerprinting to distinguish model families (Wu et al., 2025), while others attempt to recover model lineage by tracking changes in weight statistics during training or fine-tuning (Horwitz et al., 2024).

# 7. Conclusion

This paper introduces a definition of deep learning model clones and a labeled synthetic benchmark, Model-CloneBench. We further propose MCDetector, a framework-agnostic white-box method for detecting Type-1, Type-2 and Type-3 model clones by jointly analyzing structural and weight similarity. Experiments on the synthetic benchmark show that MCDetector can distinguish different clone types, while ablation results demonstrate the complementary roles of structural and weight similarity. Experiments on real-world open-source models further show that MCDetector can reveal significant similarity difference among model families and retrieve related models. Overall, this work provides a unified and interpretable framework for analyzing model reuse and derivation, with implications for provenance tracking, compliance auditing, and security analysis.

# Acknowledgements

This work was supported by National Natural Science Foundation of China under Grant 62372145, Natural Science Foundation of Zhejiang Province under Grant LZ25F020011 and Zhejiang Provincial Natural Science Foundation of China under Grant No. LQN25F020018.

# Impact Statement

This paper presents work whose goal is to advance the field of Machine Learning. There are many potential societal consequences of our work, none which we feel must be

specifically highlighted here.

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

## A. Sensitivity to the Structural Similarity Threshold

The structural similarity threshold $\tau_s$ is used to determine whether two structurally different models can be considered Type-3 clones. To evaluate the influence of this threshold, we vary $\tau_s$ from 0.6 to 0.9 and report the clone detection performance on the synthetic benchmark.

*Table 4.* Sensitivity analysis of the structural similarity threshold $\tau_s$.

| $\tau_s$ | Type-1 | | | Type-2 | | | Type-3 | | |
|---|---|---|---|---|---|---|---|---|---|
| | P | R | F1 | P | R | F1 | P | R | F1 |
| 0.6 | 1.00 | 1.00 | 1.00 | 1.00 | 1.00 | 1.00 | 1.00 | 1.00 | 1.00 |
| 0.7 | 1.00 | 1.00 | 1.00 | 1.00 | 1.00 | 1.00 | 1.00 | 0.91 | 0.96 |
| 0.8 | 1.00 | 1.00 | 1.00 | 1.00 | 1.00 | 1.00 | 1.00 | 0.38 | 0.55 |
| 0.9 | 1.00 | 1.00 | 1.00 | 1.00 | 1.00 | 1.00 | 0.00 | 0.00 | 0.00 |

The results show that Type-1 and Type-2 clones are insensitive to $\tau_s$, because they preserve identical structures by definition. In contrast, Type-3 performance is more sensitive to the threshold. When $\tau_s$ is too strict, many structurally modified Type-3 clones are rejected, leading to a substantial drop in recall. We therefore set $\tau_s = 0.7$ in the main experiments as a balanced choice.

## B. Comparison with Alternative Weight Similarity Metrics

We further compare the proposed weight similarity metric with two alternative parameter-level metrics: simple cosine similarity and Frobenius-distance-based similarity. These baselines directly compare aligned parameter tensors and are therefore more sensitive to local neuron or attention-head permutations. In contrast, MCDetector uses bidirectional maximum cosine matching, which compares parameter direction distributions without requiring exact parameter alignment.

*Table 5.* Comparison with alternative weight similarity metrics.

| Clone Type | MCDetector | | | Cosine | | | Frobenius | | |
|---|---|---|---|---|---|---|---|---|---|
| | P | R | F1 | P | R | F1 | P | R | F1 |
| Type-1 | 1.00 | 1.00 | 1.00 | 1.00 | 0.60 | 0.75 | 1.00 | 0.40 | 0.57 |
| Type-2 | 1.00 | 1.00 | 1.00 | 0.82 | 1.00 | 0.90 | 0.75 | 1.00 | 0.86 |
| Type-3 | 1.00 | 0.91 | 0.96 | 1.00 | 0.91 | 0.96 | 1.00 | 0.91 | 0.96 |

The results show that direct cosine similarity and Frobenius-based similarity are less robust for Type-1 clones under local permutation perturbations, resulting in lower recall. In contrast, MCDetector preserves perfect Type-1 and Type-2 performance while maintaining the same Type-3 performance. This suggests that bidirectional maximum cosine matching better captures parameter-distribution similarity and provides stronger robustness to local parameter reordering than direct parameter-wise comparison.

