# OpenReview forum: "An In-Depth Study on Deep Learning Model Cloning"
_ICML.cc/2026/Conference — ICML 2026 regular_

### Official Review · Reviewer_kh9u · 2026-03-10

**Soundness:** 2
**Presentation:** 3
**Significance:** 2
**Originality:** 2
**Overall Recommendation:** 3
**Confidence:** 3

**Summary:**

This paper transfers the code clone taxonomy from software engineering to deep learning models, defining four types of model clones based on structural and parameter similarity. The authors propose MCDetector, a framework agnostic method that combines structural fingerprinting via SimHash on ONNX computation graphs with weight similarity measurement using bidirectional maximum cosine statistics. The approach is evaluated on a synthetic benchmark and a collection of open source model families.

**Compliance With Llm Reviewing Policy:**

Affirmed.

**Key Questions For Authors:**

1. Could you provide an ablation comparing structural similarity alone, weight similarity alone, and the joint score on both the synthetic and real world datasets? This would clarify the marginal contribution of each component and help me better assess the core design decision.
2. The real world model families appear to have derivation relationships documented in their public model cards. Could you use this information to construct ground truth labels and report quantitative detection or retrieval metrics? This would help me better evaluate the practical effectiveness of the method.
3. Could you clarify the intended role of Type-4 clones? If detecting them is out of scope for the proposed method, an explicit discussion of this boundary would help me better understand the scope of the contribution.

**Limitations:**

See weaknesses 2 and 3.

**Strengths And Weaknesses:**

Strengths:
1. The four-type model clone taxonomy, drawn from code clone literature, gives a clear conceptual framework for model similarity and reuse.
2. Using ONNX as a unified intermediate representation makes the method applicable across frameworks without format conversion overhead.
3. Combining structural and weight similarity is a well motivated design, and the analysis of how fine-tuning, continual fine-tuning, and merging affect similarity is informative.

Weaknesses:
1. No ablation or comparison with related model similarity methods. The weight similarity component directly adopts an existing metric (such as DOCS), and several other model similarity approaches are discussed in the related work, yet none are evaluated on the same tasks. Without at least an ablation comparing structural similarity alone, weight similarity alone, and the joint score, it is difficult to assess whether combining both perspectives provides meaningful improvement over either component individually.
2. Gap between experimental evidence and claims. The synthetic benchmark contains 208 clone pairs built from small architectural components, where achieving near perfect detection for structurally identical clones is expected given the controlled construction. The real world evaluation provides qualitative analysis through heatmaps and violin plots but no quantitative metrics. Yet the abstract claim the method supports model lineage analysis, model retrieval, and intellectual property protection, none of which are experimentally validated.
3. Type-4 model clones are formally defined but absent from evaluation. The proposed method relies on structural and weight comparison, which appears fundamentally unable to detect Type-4 clones (structurally different but functionally equivalent models). This limitation is not discussed, leaving readers uncertain about the actual scope of the contribution.

---

> ### Author Rebuttal · Authors · 2026-03-31
>
> Thank you very much for taking the time to evaluate our paper and provide valuable feedback.
>
> ## Answer to Q1
>
> Thank you for your careful review and suggestions. According to the definition of clone classification, the method in this paper first distinguishes whether a model pair is Type 3 or Type 1/2 based on structural similarity, and then further differentiates between Type 1 and Type 2 through parameter similarity comparison. In our response to Question 3, we provided a table to illustrate these definitions more clearly. This demonstrates that the components of the method are interdependent and cannot be used separately.
>
> Following the your suggestion, we have supplemented an appropriate comparative experiment. During the parameter comparison stage, we applied the bidirectional cosine similarity, conventional cosine similarity, and the Frobenius norm algorithm respectively. The results indicate that the method proposed in this paper can accurately distinguish between Type 1 and Type 2 clones. In contrast, the conventional cosine similarity and Frobenius norm algorithms misclassify some Type 1 cases as Type 2, demonstrating that our method effectively addresses the "neuron permutation symmetry" issue mentioned by Reviewer-s1jg.
>
> | Approach | Our Method | | | Simple Cosine Similarity | | | Frobenius | | |
> |:---:|:---:|:---:|:---:|:---:|:---:|:---:|:---:|:---:|:---:|
> | | P | R | F1 | P | R | F1 | P | R | F1 |
> | Type 1 | 1.00 | 1.00 | 1.00 | 1.00 | 0.60 | 0.75 | 1.00 | 0.40 | 0.57 |
> | Type 2 | 1.00 | 1.00 | 1.00 | 0.82 | 1.00 | 0.90 | 0.75 | 1.00 | 0.86 |
> | Type 3 | 1.00 | 0.91 | 0.96 | 1.00 | 0.91 | 0.96 | 1.00 | 0.91 | 0.96 |
>
> ## Answer to Q2
>
> Through analysis, it can be observed that the model cards on Hugging Face and the classification framework in our paper are not aligned. While Hugging Face model cards may indicate whether a model is based on a certain base model, they do not clearly define the typological relationships between models. For instance, multiple models may be trained on the same base model, but whether they differ at the parameter or architectural level, or whether they share certain functional similarities, is not explicitly categorized in the model cards.
>
> Hugging Face model cards label relationships such as Adapter, Finetune, Merge, and Quantization, but these labels do not correspond directly to the cloning types defined in this paper. As a result, it is not feasible to measure them using specific metrics. Therefore, we have chosen to present the experimental results in the form of Figure 5.
>
> ## Answer to Q3
>
> This paper draws on the classification concepts from the field of code clone detection and proposes model clone classification. The specific classification is shown in the following table:
>
> | Criterion | Type-1 | Type-2 | Type-3 | Type-4 |
> |-----------|--------|--------|--------|--------|
> | Structural Similarity | = 1 | = 1 | > $\theta$ | — |
> | Weight Similarity | = 1 | < 1 | < 1 | — |
> | Functional Similarity | — | — | — |  > $\theta$ |
>
> Where $\theta$ represents the similarity threshold. According to the classification definition, Type 1/2/3 clones can be referred to as representation clones, while Type 4 clones can be referred to as behavioral clones. In the field of code cloning, Type 4 clones are code segments that are completely different in text but functionally similar. For example, bubble sort and quick sort code have low textual similarity but similar functionality. Such clones are typically studied as separate research topics in software engineering community. In the field of model cloning, if two models have different neural network architectures. For instance, one is an RNN-based model and the other is a Transformer-based model, but both perform the task of translating from English to Spanish, according to the definition in this paper, these two models can be identified as Type 4 model clones. This paper establishes the entire classification system to lay a benchmark for future work. Therefore, within the scope of this paper, detection research on Type 4 clones is not conducted.
>
> Thank you very much for your time and consideration. We would be very grateful if this paper could receive your recognition.We will be sure to include the above points in our revision to improve and clarify the motivation of our paper.

---

> > ### Author Rebuttal · Reviewer_kh9u · 2026-04-01
> >
> > Thank you for the detailed responses. Q3 regarding Type-4 scope is now clear and I consider it resolved.
> >
> > However, I have follow-up questions on Q1 and Q2.
> >
> > Q1 about the ablation: The comparison between SDOCS, cosine similarity, and Frobenius norm addresses which weight metric is best, but does not answer my original question. I asked whether combining structural and weight similarity provides meaningful improvement over either component alone. Specifically: what happens if you use only structural similarity (SimHash) for clone detection, or only weight similarity (SDOCS)? The claim that the two components "are interdependent and cannot be used separately" is itself a testable hypothesis. Even if the pipeline is sequential, reporting detection performance at each stage would clarify where the method's discriminative power actually comes from.
> >
> > Q2 about the real-world quantitative evaluation: I understand that HuggingFace model card labels do not map directly to your clone types. However, the base_model field and documented derivation chains do provide ground truth for whether two models share lineage. A retrieval experiment (given a query model, can the method rank its known derivatives higher than unrelated models?) would be feasible without requiring exact type labels and would provide quantitative evidence that is currently missing from the real-world evaluation.

---

> > > ### Author Response · Authors · 2026-04-02
> > >
> > > Thank you very much for your further clarification. In response to your question, we provide the following reply:
> > >
> > > # Answer to Q1：
> > >
> > > Following your suggestion, we have added an ablation study on the synthetic dataset to demonstrate the respective discriminative roles of structure similarity and weight similarity. The results are shown in the table below:
> > >
> > >
> > > | Approach                      | Our Method ||| only structural similarity ||| only weight similarity |||
> > > |:---:|:---:|:---:|:---:|:---:|:---:|:---:|:---:|:---:|:---:|
> > > |                               | P          | R          | F1         | P          | R        | F1       | P    | R  | F1 |
> > > | Type 1                         | 1.00     | 1.00     | 1.00     | -          | -        | -        | - | - | - |
> > > | Type 2                         | 1.00     | 1.00     | 1.00     | -          | -        | -        | -      | -  | -   |
> > > | Type 3                         | 1.00     | 0.91     | 0.96     | 1.00     | 0.91   | 0.96   | -      | -  | -   |
> > >
> > >
> > > The results show that using only structure similarity can effectively identify Type-3, but fails to distinguish between Type-1 and Type-2, because by definition the structures of these two types remain identical. Using only weight similarity cannot independently distinguish Type-1, Type-2, or Type-3; even when the parameters are exactly the same, we lack sufficient theoretical basis to prove that it is a Type-1 clone. In contrast, only by combining structure and weight information can our method achieve the overall discrimination of Type-1, Type-2, and Type-3. Furthermore, through this ablation study we can demonstrate that structure similarity and weight similarity are not redundant designs in our method, but rather play complementary discriminative roles: structure information is primarily used to identify Type-3, while weight information is used to further distinguish between Type-1 and Type-2 under the condition of identical structure. Only by combining the two can we fully support the clone taxonomy proposed in this paper.
> > >
> > >
> > > # Answer to Q2：
> > >
> > > To address the reviewer's question regarding "real-world quantitative evaluation", we constructed a ground truth dataset based on evolutionary relationships using the `base_model` field provided in HuggingFace model cards and their documented derivation chains. The evolutionary relationship of each model provides clear upstream and downstream connections for its derived models, without relying on specific clone type labels. On this basis, we selected each model as a query and performed retrieval from the dataset using our method to evaluate the similarity between the model and its known derived models. The goal of the experiment was to retrieve a model's known derived models and to see whether our retrieval method could prioritize returning relevant models within the model dataset, thereby validating the effectiveness of our method. The experimental results are shown in the following table:
> > >
> > > | Approach      | Hit@1 | Hit@2 | Hit@3 |
> > > |---------------|-------|-------|-------|
> > > | MCDetector    | 0.58  | 0.92  | 1     |
> > >
> > > We computed the hit rates for Top-1 (hit@1), Top-2 (hit@2), and Top-3 (hit@3) in retrieval. The results show that hit@1 is 0.58, hit@2 is 0.92, and hit@3 is 1. This means that in 58% of the cases, the most relevant model (based on structure and weight similarity) is correctly returned as the derived model of the query model. Moreover, the experimental result achieves a perfect hit@3 of 1, meaning that within the top‑3 ranked models, we are able to return a known derived model in every case.
> > >
> > > Upon inspection, we found that the hit@1 value is relatively low. We examined the relevant HuggingFace repositories and discovered that for some models, the `base_model` underwent significant fine‑tuning or other changes after release, which leads to low weight similarity between these models and their derived models. Since our method considers both structure and weight similarity, the low weight similarity affects the accuracy of hit@1.
> > >
> > > Our retrieval method combines both structure and weight similarity. When facing some models with low weight similarity, structure similarity can still help identify the derivation relationship. This is why we achieve a perfect result at hit@3. Through this experiment, we demonstrate that even when some models have low weight similarity, our combined approach that incorporates structure similarity can still accurately find the derivation relationships of the models. This proves the effectiveness of our method in real‑world applications.
> > >
> > > Thank you very much for your careful review. We hope this response fully addresses your questions. Once again, thank you for your suggestions, and we hope that this paper will ultimately receive your approval.

---

### Official Review · Reviewer_SCRF · 2026-03-10

**Soundness:** 2
**Presentation:** 3
**Significance:** 2
**Originality:** 3
**Overall Recommendation:** 4
**Confidence:** 3

**Summary:**

This paper addresses the intellectual property and security risks associated with deep learning model reuse by proposing the concept of "model cloning" and defining four types. A framework-independent model clone detection method, MCDetector, is designed. The method quantifies model similarity through unified computational graph representation, structural similarity analysis, and weighted similarity analysis. Its effectiveness is validated on synthetic datasets and real open-source models, providing a foundation for model lineage analysis, retrieval, and intellectual property protection.

**Compliance With Llm Reviewing Policy:**

Affirmed.

**Final Justification:**

The authors' rebuttal is good for addressing my main concern. But considering the novelty of this work, I will maintain the original score.

**Key Questions For Authors:**

(1) What is the difference between “model cloning” and “model lineage” described in the article, and how can we distinguish whether they are the same type of task? The concept of “model cloning” and the four classification definitions proposed in the article highly overlap with model lineage research. Is it necessary to define them separately?

**Limitations:**

yes

**Strengths And Weaknesses:**

Strengths:

+The structure is clear, the description is clear, and the language is fluent.

+ Model cloning in the field of deep learning is proposed based on code cloning in software engineering.

+ The experimental datasets take into account both self-built synthetic datasets and real open-source models.

+ The method described in this paper enhances its versatility across different framework models by converting models from different architectures and sources into the ONNX format.

Weaknesses：
- The experiments mentioned in the paper only used the proposed method and were not compared with other methods, even for basic similarity comparisons. Therefore, the method in the paper achieves similarity comparison of model clones by comparing the computational structure and parameter distribution of the models. Given this increased complexity, is the performance improvement compared to ordinary similarity calculation methods or other methods meaningful?

- The method in the paper uses a manually designed method for extracting computational structural features and parameter distribution features. How does this method compare to methods based on model representation? No relevant experiments or descriptions are available.

- The method mentioned in the paper requires manipulation of the computational structure and parameters in the model, which requires all models to be white-box models. Model privacy has not been considered.

- The paper proposes four categories of model cloning, but only the results for the first three categories are observed in Table 1. There is no relevant content for the fourth category.

- The experimental section of the paper does not provide specific model similarity calculation results or descriptions.

- In RQ1, is there any basis for setting the structural similarity threshold for Type-3 clones to 0.7?

---

> ### Author Rebuttal · Authors · 2026-03-31
>
> Thank you very much for your positive evaluation of our paper. We also highly value the issues you raised. Here are our responses:
>
> ## Answer to Q1
>
> Thank you for your thorough review. We understand that the model lineage emphasizes the historical parent-child evolutionary relationship, whereas model cloning emphasizes the current similarity in structure, parameters, or functionality, as well as the categorical relationship between the two models. The former answers "where the model comes from," while the latter answers "how similar the two models are and what type of reuse relationship they fall into." Therefore, models with an inheritance relationship are highly likely to exhibit significant similarity and can be classified as model clones. However, two models that are clones do not necessarily evolve from the same model family; the two concepts are not equivalent. We will clarify this point more explicitly in the revised manuscript.
>
> ## Answer to Weaknesses1
>
> Regarding the comparative experiment, there are currently no methods or works in this field that share the same objective as this paper, which is why we did not compare with other approaches. Following the your suggestion, we have supplemented an appropriate comparative experiment. During the parameter comparison stage, we applied the bidirectional cosine similarity, conventional cosine similarity, and the Frobenius norm algorithm respectively. The results indicate that the method proposed in this paper can accurately distinguish between Type 1 and Type 2 clones. In contrast, the conventional cosine similarity and Frobenius norm algorithms misclassify some Type 1 cases as Type 2, demonstrating that our method effectively addresses the "neuron permutation symmetry" issue mentioned by Reviewer-s1jg.
>
> | Approach | Our Method | | | Simple Cosine Similarity | | | Frobenius | | |
> |:---:|:---:|:---:|:---:|:---:|:---:|:---:|:---:|:---:|:---:|
> | | P | R | F1 | P | R | F1 | P | R | F1 |
> | Type 1 | 1.00 | 1.00 | 1.00 | 1.00 | 0.60 | 0.75 | 1.00 | 0.40 | 0.57 |
> | Type 2 | 1.00 | 1.00 | 1.00 | 0.82 | 1.00 | 0.90 | 0.75 | 1.00 | 0.86 |
> | Type 3 | 1.00 | 0.91 | 0.96 | 1.00 | 0.91 | 0.96 | 1.00 | 0.91 | 0.96 |
>
> ## Answer to Weaknesses2
>
> The reviewer mentioned in the second point of the weaknesses that a model representation method could be used for comparison, which is indeed a potentially good approach.
>
> As this paper is an initial effort to classify and explore the related concepts, we opted for traditional methods that offer better interpretability, and we hope you will acknowledge this choice. We will continue to explore further in our subsequent work.
>
> ## Answer to Weaknesses3
>
> The method proposed in this paper specifically targets white-box models. The cloning types we define are categorized based on internal model content comparison. Therefore, to achieve high-precision identification of cloning types (Type 1–3), in-depth analysis of parameter distributions is essential. For assessing the similarity of black-box models, methods such as model watermarking or fingerprinting are typically employed. These two directions are complementary. We will add a "Limitations" discussion in the revised manuscript to clarify that this method is not directly applicable to encrypted models or black-box API services.
>
> ## Answer to Weaknesses4
>
> Due to the character limitaion. Please refer to our response to Reviewer mo5w for the discussion on Type 4 clones (Answer to Q2). Thank you!
>
> ## Answer to Weaknesses5
> We are currently analyzing the similarity features of the relevant models and do not yet have significant findings to report. We will supplement the revised manuscript with relevant content accordingly. Thank you.
>
> ## Answer to Weaknesses6
>
> Regarding the issue of **structural similarity threshold**: if the threshold is set too small, even modules like the encoder and decoder in a Transformer, which have similar structures, might cause different types of models to be incorrectly defined as clones. Conversely, if the threshold is set too large, the overly strict condition may prevent models that only differ by the removal of unnecessary modules from being correctly classified as clones. We conducted experiments on this issue, and the results show that when the threshold is too small, the algorithm identifies all models in the dataset as clones; when the threshold is too large, the algorithm classifies all models as non-clones.
>
> | Threshold | Type 1 | | |Type 2 | | | Type 3 | | |
> |:---:|:---:|:---:|:---:|:---:|:---:|:---:|:---:|:---:|:---:|
> | Metric | P | R | F1 | P | R | F1 | P | R | F1 |
> | 0.6 | 1.00 | 1.00 | 1.00 | 1.00 | 1.00 | 1.00 | 1.00 | 1.00 | 1.00 |
> | 0.7 | 1.00 | 1.00 | 1.00 | 1.00 | 1.00 | 1.00 | 1.00 | 0.91 | 0.96 |
> | 0.8 | 1.00 | 1.00 | 1.00 | 1.00 | 1.00 | 1.00 | 1.00 | 0.38 | 0.55 |
> | 0.9 | 1.00 | 1.00 | 1.00 | 1.00 | 1.00 | 1.00 | 0 | 0 | 0 |
> ---

---

> > ### Author Rebuttal · Reviewer_SCRF · 2026-04-03
> >
> > Thanks for the rebuttal.

---

> > > ### Author Response · Authors · 2026-04-04
> > >
> > > Thank you for your timely response and for recognizing our rebuttal.
> > >
> > > We sincerely hope our detailed responses have helped strengthen your confidence in this work and your overall review assessment. Please feel free to let us know if you have any remaining questions or concerns, and we will be glad to provide further clarification.

---

### Official Review · Reviewer_s1jg · 2026-03-11

**Soundness:** 3
**Presentation:** 4
**Significance:** 2
**Originality:** 2
**Overall Recommendation:** 4
**Confidence:** 4

**Summary:**

This paper proposes a method for comparing deep learning models using a combination of topological similarity (via SimHash features on normalized computation graphs) and weight similarity (using tensor sets and a variant of cosine distance). The work has applications in automated detection of different model clones (as defined by the authors) and potentially identifying model reuse and provenance. The authors release a small dataset (n=208) of model clones that are classified according to clone type. The authors show promising performance of their method in discriminating both the synthetic clones in their dataset as well as models "in the wild" posted on the HuggingFace repository.

**Compliance With Llm Reviewing Policy:**

Affirmed.

**Key Questions For Authors:**

(1) Do the authors have additional evidence that the practice of model cloning is widespread and/or that the usage of semi-automated methods for clone detection would provide appreciable utility in large scale repositories? While the argument about licensing (section 1) is convincing, figure 5 does not seem to support the idea that models are routinely cloned outside of explicitly specified model famlies.

(2) Do the authors have additional motivation for the definition of clone type IV? Careful definition of "functionality" is needed in this case and this ambiguity does not seem to be explored in the synthetic ModelCloneBench dataset or the Huggingface experiments.

(3) Do the authors have statistics on how often the UNK parameter tag (section 3.1) is needed during experiments?

(4) Do the authors intend to provide a more disciplined performance metric within the Huggingface experiments? Given that performance on the synthetic dataset is nearly perfect, this stakes a strong claim against the performance of the method (that does not seem to be fully validated). The relationships in the Huggingface models are qualitatively implied, but the paper would benefit from an explicit taxonomy and performance assessment beyond figure 5.

(5) As an extension to question (4), do the authors intend to provide additional threshold values and how they correspond to different qualitative factors of model inheritance (such as the ones identified in the Huggingface experiments - reuse, fine-tuning, and derivation)? The paper would benefit from demonstrated interpretability of the metric and its alignment with these qualitative relationships. For example, if two different model families show asymmetric swings in this metric between their 2.5b and 7b parameter variants, it makes the metric harder to interpret. This would also help to answer questions such as how can one identify two models that are the same except for permutation of the transformer heads?

**Limitations:**

Yes

**Strengths And Weaknesses:**

The paper has a moderate degree of technical soundness - while the methods employed are consistent and correct, they lack completeness in certain areas. For example, the paper would benefit from a more disciplined approach to exploring issues like neuron permutation symmetry or specific architectural choices like max-pooling.

The paper has a high degree of presentation. Concepts are well-explained and built up from sufficiently elementary background. The enumeration of model clone types and the various challenges in clone detection are easy to follow.

The paper has low to moderate demonstrated significance. While the potential implications of model clone proliferation are elucidated, the hypothesis that this is common enough to be a practical problem is not demonstrated beyond limited examples using model naming (which the author's agree is a poor proxy for the underlying problem).

The paper has moderate originality. Several existing methodologies (SimHash, common substring matching, SDOCS) are combined in a reasonable way to compare models. The significance of these methods in the context of common issues in model clone detection (such as permutation symmetry, large computation cost) is presented. This represents a reasonably novel application of these methods in a way tailored to the model clone detection problem.

---

> ### Author Rebuttal · Authors · 2026-03-31
>
> Thank you very much for your positive evaluation of our paper. We are glad you found it well-presented. We also highly value the issues you raised. Here are our responses:
>
> ## Answer to Q1
> Model cloning is a concept that encompasses a broad scope. When a model is fine-tuned based on another model, the two can be considered clones. Quantizing a model to create a new version can also result in a model clone. Even connecting new network layers to the output of a base model to form a new model can give rise to a model clone.  By searching keywords such as model fine-tuning, model quantization, and model merging on DBLP, we retrieved 1,476 related papers. This indirectly demonstrates the practical existence of model cloning.
>
> Figure 5 indeed does not indicate whether a model is cloned from a specific family. According to related research in software engineering, determining explicit cloning relationships between software not only requires assessing similarity but also involves multidimensional information such as licenses, release timelines, and developers. This issue could also be explored as an independent research topic.
>
> ## Answer to Q2
>
> Due to the character limitaion. Please refer to our response to Reviewer mo5w for the discussion on Type 4 clones. Thank you!
>
> ## Answer to Q3
>
> When models are built using non-standard custom operators, information loss may occur during ONNX export. Therefore, to better ensure the consistency of sequential structures in the models being compared, we have introduced this strategy. The datasets used in this paper consist of models defined in a standardized manner within ONNX, and our statistics show that no "unknown" labels appeared in the experiments. However, we believe this strategy can indeed effectively address some real-world scenarios. We hope the reviewers will acknowledge our approach. We plan to inject non-standard operators into the synthetic dataset in the revised version and optimize our experiments accordingly.
>
> ## Answer to Q4
>
> Through analysis, it can be observed that the model cards on Hugging Face and the classification system in our paper are not aligned. While Hugging Face model cards may indicate whether a model is based on a certain base model, they do not clearly define the typological relationships between models. For instance, multiple models may be trained on the same base model, but whether they differ at the parameter or architectural level, or whether they share certain functional similarities, is not explicitly categorized in the model cards.
>
> Hugging Face model cards label relationships such as Adapter, Finetune, Merge, and Quantization, but these labels do not correspond directly to the cloning types defined in this paper. As a result, it is not feasible to measure them using specific metrics. Therefore, we have chosen to present the experimental results in the form of Figure 5.
>
> ## Answer to Q5
>
> Thank you for your careful review and for pointing out a highly valuable research direction. Currently, our Type 2 classification is relatively coarse-grained, encompassing all cases of fine-tuning, quantization, and similar operations under this category. The existing parameter threshold cannot reflect the inheritance relationship between models; it can only indicate whether a similarity exists, but not deduce inheritance based on the threshold. We will conduct experiments in the near future to explore whether statistical patterns exist in parameter similarity for Type 2 clones such as fine-tuning and quantization. Relevant findings will be added to the published version in appropriate sections.
>
> Regarding the issue of identical model structures with perturbed attention heads that you mentioned, the method proposed in this paper employs the bidirectional cosine similarity algorithm (SDOCS), which can effectively identify such model clones. To demonstrate the effectiveness of this method, we conducted a comparative experiment: when comparing parameter similarity, we applied bidirectional cosine similarity, conventional cosine similarity, and the Frobenius norm algorithm respectively. The results show that our method accurately distinguishes between Type 1 and Type 2 clones, whereas conventional cosine similarity and the Frobenius norm algorithm misclassify some Type 1 cases as Type 2. In other words, they cannot effectively address the issue of attention head perturbation mentioned by the reviewer.
>
> | Approach | Our Method | | | Simple Cosine Similarity | | | Frobenius | | |
> |:---:|:---:|:---:|:---:|:---:|:---:|:---:|:---:|:---:|:---:|
> | | P | R | F1 | P | R | F1 | P | R | F1 |
> | Type 1 | 1.00 | 1.00 | 1.00 | 1.00 | 0.60 | 0.75 | 1.00 | 0.40 | 0.57 |
> | Type 2 | 1.00 | 1.00 | 1.00 | 0.82 | 1.00 | 0.90 | 0.75 | 1.00 | 0.86 |
> | Type 3 | 1.00 | 0.91 | 0.96 | 1.00 | 0.91 | 0.96 | 1.00 | 0.91 | 0.96 |
> ---
>
> We will be sure to include the above points in our revision to improve and clarify the motivation of our paper.

---

### Official Review · Reviewer_mo5w · 2026-03-18

**Soundness:** 3
**Presentation:** 3
**Significance:** 3
**Originality:** 3
**Overall Recommendation:** 4
**Confidence:** 3

**Summary:**

The article focuses on detecting deep learning model clones i.e. models derived through reuse, fine-tuning, compression, or modification, which raise concerns about intellectual property, provenance, and security risks. It introduces MCDetector, a framework that identifies clones using structural similarity (via ONNX graphs and SimHash fingerprints) and weight similarity (using the SDOCS metric). The approach defines four clone types and is evaluated on both synthetic (ModelCloneBench) and real-world HuggingFace models. It addresses deep learning model reuse and IP protection, introduces a clear clone taxonomy, combines structural and parameter similarity for detection, uses an ONNX-based framework-agnostic design, and validates results on synthetic and real-world datasets while revealing model evolution patterns.

**Compliance With Llm Reviewing Policy:**

Affirmed.

**Key Questions For Authors:**

The work addresses an important problem and presents a practical framework with a useful clone taxonomy, showing strong results for Typen1 and Typen2.However, it raises few concerns that need to be addressed:
1. The algorithmic novelty is limited, as the approach largely combines existing methods like SimHash and cosine similarity.
2. It seems that xperimental evaluation is incomplete: Typen4 clones are not properly addressed, and their functional similarity definition lacks clarity and a detection method.
3. The evaluation setup is somewhat weak, with synthetic data not fully reflecting real-world complexity and no scalability analysis for large models (e.g., LLMs).
4. Lastly, dependence on ONNX may limit applicability, and the impact statement does not sufficiently discuss broader societal implications.

**Limitations:**

1. The authors has presnets a good idea however it showcase limited novelty (relies on SimHash, cosine similarity); incomplete evaluation (only Typen1–3 tested, Typen4 unclear and lacks detection).
2. The synthetic data used for analysis may not reflect real-world complexity; no scalability analysis for large models (e.g., LLMs).
3. Similarly, ONNX dependency may restrict applicability; weak impact statement with little discussion of societal implications.

**Strengths And Weaknesses:**

1. The paper incorporates deep learning model reuse and IP protection; introduces a clear clone taxonomy.
2. It detects clones using both structural and parameter similarities in a framework-agnostic (ONNX-based) setup.
3. It proposed validation on synthetic and real-world datasets, revealing model evolution patterns.
4. The paper though has potential but needs stronger novelty, scalability analysis, clearer Typen4 handling, and deeper theoretical grounding.

---

> ### Author Rebuttal · Authors · 2026-03-31
>
> Thank you very much for your positive evaluation of our paper. We sincerely appreciate your recognition of the importance of the problem addressed in this paper. We also highly value the questions you raised, and our responses are as follows:
>
> ## Answer to Q1
>
> Regarding the novelty of the algorithm, our investigation reveals that graph similarity comparison is a computationally expensive task. Traditional graph similarity methods fail to handle even moderately large models. As indicated by the recent work "Enhancing Graph Edit Distance Computation: Stronger and Orientation-based ILP Formulations," the optimized exact graph edit distance algorithm requires several minutes for a single computation on large graphs, such as CORA (with 2,708 nodes) and PUBMED (with 19,717 nodes). Through analysis, we find that the forward propagation process of deep neural networks can typically be modeled as a directed computational graph. For feedforward networks, this graph is usually a directed acyclic graph, while for recurrent neural networks, their temporally unfolded forms can be represented as directed acyclic graphs. Therefore, we address this challenge by converting larger graph structures into linear sequential structures for comparison. We employ the SimHash algorithm for sequence similarity comparison, thereby enabling structural similarity analysis for large neural network models.
>
> Furthermore, this paper includes the following innovative contributions: it is the first to formally define deep learning model cloning as an independent problem, proposes a four-category model cloning taxonomy, and establishes a framework-agnostic detection pipeline for heterogeneous model ecosystems.
>
> ## Answer to Q2
>
> This paper draws on the classification concepts from the field of code clone detection and proposes model clone classification. The specific classification is shown in the following table:
>
> | Criterion | Type-1 | Type-2 | Type-3 | Type-4 |
> |-----------|--------|--------|--------|--------|
> | Structural Similarity | = 1 | = 1 | > $\theta$ | — |
> | Weight Similarity | = 1 | < 1 | < 1 | — |
> | Functional Similarity | — | — | — |  > $\theta$ |
>
> Where $\theta$ represents the similarity threshold. According to the classification definition, Type 1/2/3 clones can be referred to as representation clones, while Type 4 clones can be referred to as behavioral clones. In the field of code cloning, Type 4 clones are code segments that are completely different in text but functionally similar. For example, bubble sort and quick sort code have low textual similarity but similar functionality. Such clones are typically studied as separate research topics in software engineering community. In the field of model cloning, if two models have different neural network architectures. For instance, one is an RNN-based model and the other is a Transformer-based model, but both perform the task of translating from English to Spanish, according to the definition in this paper, these two models can be identified as Type 4 model clones. This paper establishes the entire classification system to lay a benchmark for future work. Therefore, within the scope of this paper, detection research on Type 4 clones is not conducted.
>
> ## Answer to Q3
>
> Thank you for your suggestions. The current synthetic dataset includes basic fully connected layers, RNN operators, LSTM operators, CNN operators, and Transformer encoder-decoder structures. We plan to further investigate and incorporate related neural network operators into the synthetic dataset, and we hope the reviewers will recognize this work and give us the opportunity to make improvements.
>
> Regarding the scalability analysis mentioned, the open-source datasets collected for experiments already include large-scale models ranging from 1B to 7B parameters. We will add clarifications on this in the revised version.
>
> ## Answer to Q4
>
> The purpose of using ONNX in this paper is to map models from different frameworks into a unified intermediate representation, thereby enabling consistent cross-framework analysis. Thank you for your guidance. This method is not universally applicable to all model forms; for models containing a large number of custom operators, those that cannot be fully exported to ONNX, or those provided only in the form of black-box APIs, the applicability of the approach proposed in this paper has its limitations. In the future, it could be extended by adopting a more abstract intermediate representation layer or by combining behavioral-level methods. We will also include this limitation in the "Threats to Validity" section.
>
> We will be sure to include the above points in our revision to improve and clarify the motivation of our paper. We hope you can recognize our efforts and grant us the opportunity. We will continue this work.

---

### Decision · Program_Chairs · 2026-04-30

**Decision:**

Accept (regular)

**Comment:**

The article proposes and takes a first step to addressing issues in model clones, creating an initial taxonomy, and basic methods to quantifying similarity between files and releasing a small dataset of clones for research purposes. While no reviewer has championed the work, I've experienced the legal uncertainty that this issue creates in corporate environments. As such, given the current scope of reviews, I will recommend this paper for acceptance as it provides a good step in tackling something that is a bigger problem than many realize.